# Whole Genomic Characterization of *Streptococcus iniae* Isolates from Barramundi (*Lates calcarifer*) and Preliminary Evidence of Cross-Protective Immunization

**DOI:** 10.3390/vaccines11091443

**Published:** 2023-08-31

**Authors:** Sunita Awate, Salma Mubarka, Roland G. Huber

**Affiliations:** 1UVAXX Pte Ltd., 203 Henderson Industrial Road, #12-01, Singapore 159546, Singapore; salma@uvaxx.com; 2Bioinformatics Institute (BII), Agency for Science, Technology and Research (A*STAR), Matrix #07-01, 30 Biopolis Street, Singapore 138671, Singapore

**Keywords:** aquaculture, bacterial pathogens, vaccination, *Lates calcarifer*, *Streptococcus iniae*

## Abstract

*Lates calcarifer*, also known as Barramundi or Asian seabass, is a highly productive and fast-growing species that is well suited to large-scale aquaculture due to its attractive harvestable yields (premium fish). This fish has been envisioned as having the potential to be the “Salmon of Tropics”. Cultivating *Lates calcarifer* in aquaculture poses challenges, as the dense populations that make such aquaculture commercially viable facilitate the rapid spread of infectious diseases, which in turn significantly impact yield. Hence, the immunization of juveniles is necessary, and the development of new immunization agents enhances the efficiency of aquaculture and improves food security. In our study, we characterize seven novel strains of the bacterial pathogen *Streptococcus iniae* that were collected from commercial fish farms in Singapore and Australia. We find that the capsular operon in our strains is highly conserved and identify a number of major surface antigens previously described in Streptococcus. A genome analysis indicates that the present strains are closely related but form distinct strains within the *S. iniae* species. We then proceed to demonstrate that inoculation with the inactivated strain P3SAB cross-protects *Lates calcarifer* against *S. iniae* infections in vivo from a variety of strains found in both Singapore and Australia.

## 1. Introduction

According to the United Nations Food and Agriculture Organization (FAO), current aquaculture produces an approximately equal share of global seafood production as wild catch. Significant growth in the aquaculture market suggests that aquaculture will surpass wild catch as a source of fish in the coming years [1]. A separate FAO report indicates that the consumption of fish accounts for approximately 20% of daily protein intake globally, making fish, and therefore aquaculture, a key backbone of human nutrition [2] and global food security. Over 90% of cultured fish by mass and approximately 85% of cultured fish by value is cultivated in the Asia-Pacific region [3].

*Lates calcarifer*, also known as Barramundi or Asian seabass [4], is a fish of major commercial interest with annual aquaculture production in excess of 30,000 metric tons. *Lates calcarifer* is found in costal water and freshwater from the Indian coast to the southwestern Pacific, generally in warmer waters [5]. This fish can be cultured in seawater, brackish water, and freshwater but spawning generally occurs in brackish water near the mouths of rivers. The fish is fast growing and popular for human consumption, which explains the interest in this species for aquaculture enterprises. Initially, farming of *Lates calcarifer* started in Thailand and has since expanded throughout Southeast Asia and to northern Australia [5]. Dense stocking in aquaculture enclosures increases the likelihood and impact of infectious diseases. *Lates calcarifer* is susceptible to several emerging viral (*Megalocytivirus*, *Lates calcarifer herpesvirus*, and *Lates calcarifer Birnavirus*) and bacterial infections (*Tenacibaculum maritimum*, *Vibrio harveyi*, and *Streptococcus iniae*) [6,7,8,9,10]. Infection by *Streptococcus* bacteria can cause high mortality rates over a period of 3–7 days, but chronic outbreaks lasting weeks with continuing mortality are possible in both marine and freshwater fish. High water temperatures in tropical and subtropical regions are particularly conducive to the spread of this Gram-positive bacterium. Specifically, in 1999 Bromage et al. [10] identified *S. iniae* as a significant pathogen threatening *Lates calcarifer* populations in Australian aquaculture. The isolated strains were shown to induce mortality in 60–80% of sample populations. Clinical signs in infected fish include unilateral or bilateral exophthalmia, corneal opacity, hemorrhage at the base of fins, skin discoloration, tail rot, and erratic swimming behavior. The bacterium can be isolated from the brains of surviving fish, indicating that a reservoir of *S. iniae* persists even after an outbreak and the removal of deceased fish. Furthermore, *S. iniae* has also been isolated in the aftermath of major disease-induced mortality from other commercially relevant fish species, e.g., tilapia or rabbit fish. Hence, the prevalence of *S. iniae* constitutes a major threat to the successful cultivation of *Lates calcarifer*.

In addition to the threat, *S. iniae* poses some concern for human health as well. While uncommon, cases of *S. iniae* infections have been documented in humans, starting in 1995 in Canada [11] with more recent cases also occurring in Asia, e.g., in Hong Kong [12] and Singapore [13]. Symptoms from *S. iniae* infection in humans can include bacteremia cellulitis, arthritis, meningitis, and endocarditis [13].

A number of proteins were previously identified for their suitability as vaccine candidates. In particular, the protein Sip11 has been shown to be protective against *S. iniae* infection in Japanese flounder [14]. Formalin-killed bacteria have also been explored as a vaccine candidate in rainbow trout and have proven effective in reducing mortality [15].

In this study, we investigate the genetics of seven independent isolates of *S. iniae* obtained from infected *Lates calcarifer* and *Eleutheronema tetradactylum* cultured in Singapore and Australia. We used long-read nanopore sequencing [16] to obtain and assemble the individual *S. iniae* genomes and analyze major differences and similarities, in particular in the capsular operon of *S. iniae*, which is a key protective antigen and host defense mechanism [17] and is important for virulence [18].

Subsequently, we demonstrate that broadly neutralizing vaccines can be obtained from the inactivated P3SAB strain of *S. iniae* that shows high cross-protection against both Singaporean and Australian strains in *Lates calcarifer*.

## 2. Materials and Methods

### 2.1. Isolation of S. iniae Strains

Seven *S. iniae* strains were isolated during disease outbreaks from various farms in Singapore and Australia (Table 1). Out of the seven strains, six were isolated from Asian seabass (*Lates calcarifer*) and one strain was isolated from a threadfin fish (*E. tetradactylum*).

### 2.2. Identification of S. iniae by PCR

PCR identification of *S. iniae* was performed using specific primers based on the lactate oxidase-encoding (lox) gene [14]. Briefly, colonies were sub-cultured in tryptic soy broth (TSB) medium (22092, Sigma-Aldrich, St. Louis, MO, USA) and incubated overnight at 30 °C. TSB culture was boiled for 5 min at 95 °C and centrifuged to collect supernatant for PCR analysis. An 870 bp fragment of the lox gene was amplified in a PCR reaction containing 1 μM of each primer (LOX-1: 5′ AAGGGGAAATCGCAAGTGCC-3′and LOX-2: 5′-ATATCTGATTGGGCCGTCTAA-3′), 0.2 mM of each dNTP, 2.5 U DNA polymerase (PL1204, Vivantis, Shah Alam, Malaysia), and 1× reaction buffer along with the boiled suspension as the template. Thermocycling was carried out using an initial denaturation step at 95 °C for 1 min followed by 35 cycles of denaturation at 92 °C for 1 min, annealing at 60 °C for 1 min, extension at 72 °C for 2 min, and a final extension at 72 °C for 5 min. Positive and negative controls were included in each PCR reaction. PCR products were detected using 1% agarose gel electrophoresis containing 1× safe green dye (SD0101, Vivantis, Shah Alam, Malaysia). Bands were visualized under blue light transilluminator (TT-BLT-470, Hercuvan, Shah Alam, Malaysia). Post confirmation, *S. iniae* strains were stored in 20% glycerol at −80 °C for later use.

### 2.3. Genomic DNA Extraction

*S. iniae* isolates were cultured for approximately 7–8 h in 50 mL TSB until mid-log phase (OD600 = 0.5). The culture stock (8 mL) pellets were used to carry out gDNA extraction by phenol chloroform isoamyl alcohol (PCI) method with modification to remove capsular polysaccharides. [19] Briefly, the aqueous phase containing DNA was treated with water-saturated diethyl ether (296082, Sigma-Aldrich, St. Louis, MO, USA) and 5M sodium chloride (S-5150, Sigma-Aldrich, St. Louis, MO, USA) before precipitation with ethanol. DNA pellets were dissolved in 10 mM Tris buffer (pH 8.0) and stored at 4 °C before sequencing.

### 2.4. Library Preparation

An amount of 1.5 to 2 µg of isolated DNA was sheared to a target size of 30 kb on the Megaruptor 3 (Diagenode) according to the manufacturer’s instructions. Sheared DNA size was assessed using the Genomic Screentape on the Agilent 4200 Tapestation (Aglient Technologies, Santa Clara, CA, USA) based on the manufacturer’s instructions. An additional 0.4× Ampure XP (Beckman Coulter) size selection was performed on selected samples exhibiting increased proportion of DNA sizes <4 kb on the genomic screentape profile to aid in removal of DNA fragments <2 kb.

An amount of 500 ng of sheared and size-selected DNA was input to the PCR-free multiplexed whole-genome sequencing (WGS) library using the ligation sequencing kit (SQK-LSK109) in combination with native barcoding expansion kits (EXP-NBD104 and 114), according to the manufacturer’s instructions (Oxford Nanopore Technologies, ONT, Oxford, UK). The final barcoded library pool was loaded on a FLO-MIN106D, R9.4.1 flow cell and sequenced on the GridION (Oxford Nanopore Technologies) with super accurate (SUP) live basecalling on Guppy 5.0.12, MinKNOW release 21.05.12. Demultiplexing of basecalled, pass-filter fastq was completed using qcat (Oxford Nanopore Technologies). Two sequencing runs were conducted for sample Si 1-19-P3SAB (flow cell ID FAQ65824) and sample Si 6-21-CB Si-1 (flow cell ID FAQ94395). Sequencing parameters and results are given in Table 2.

### 2.5. Genome Assembly and Annotation

Genomes were assembled using the Flye Pipeline [20,21] specifying a circular genome. For all 7 samples, the largest fragment obtained was 2.1 Mb in length, which is consistent with previous assemblies of *S. iniae* [22]. Mean coverage ranged from 351× to 2749×. Assembly statistics are summarized in Table 2. After assembly, medaka (https://github.com/nanoporetech/medaka (accessed on 23 April 2023)) was used for genome polishing as currently recommended for Oxford Nanopore assemblies. Subsequent to assembly we used the NCBI Prokaryotic Genome Annotation Pipeline (PGAP) [23] to identify genes, which returned approximately 2000 genes per assembly, of which about 1500 genes were identified as protein-coding genes. These numbers are also consistent with previously obtained *S. iniae* annotations. Details on number and type of genes identified are shown in Table 2. A general overview of annotated genes is given in Figure 1a, with forward-strand genes highlighted in dark blue and reverse-strand genes highlighted in light blue. GC content was approximately 37% with local peaks of up to 65%. Local GC content is shown as a magenta trace in Figure 1.

### 2.6. Genome Analysis

The assembled genomes were analyzed using OrthoFinder [24,25] to identify orthologous genes and derive a phylogenetic tree considering the entire coding genome. OrthoFinder uses BLAST [26] to group orthologous genes into so-called orthogroups and subsequently provides a sequence alignment, a variety of statistics, and phylogenetic inference using the STAG approach [27]. The capsular proteins were identified by sequence alignment against the 21 kb reference capsular operon identified by Millard et al. [18] (GenBank AY904444) using MAFFT [28].

### 2.7. Vaccine Preparation

The vaccine for *S. iniae* strain P3SAB was prepared as ready-to-use water-in-oil emulsions for intraperitoneal injection. To begin with, the *S. iniae* P3SAB strain was cultured in TSB media (50 mL) and incubated overnight at 28 °C with gentle agitation (150 rpm). The culture was inactivated using formalin (252549, Sigma Aldrich, St. Louis, MO, USA) with a final concentration of 0.4% (*v*/*v*) for at least 48 h at 4–8 °C. Inactivated cultures were further washed and diluted with sterile saline, then emulsified at a 30:70 ratio using MONTANIDE ISA (Seppic) oil adjuvant. Sterility of the vaccine was verified by standard procedures.

### 2.8. Experimental Fish and Husbandry

One hundred and twelve healthy Barramundi (*Lates calcarifer*) fingerlings of approximately 12 g were obtained from a fingerling supplier in Australia and were acclimatized for 2 weeks with aeration. Fish were fed to satiation twice daily with a commercial feed (Lucky Star 4, Lucky Star Holdings, Singapore). The water temperature was maintained at 30 ± 3 °C and salinity at 10 ppt. Water quality was regulated using a hang-on filter (with 3 filtering effects: physical, chemical, and biological) and daily cleaning. Ammonia, pH, and nitrate were checked regularly and water changes (10–20%) were performed as required. Fish were screened for general pathogens before the experiment. All the experimental procedures were performed according to National Advisory Committee for Laboratory Animal Research (NACLAR) guidelines and approved by the Institutional Animal Care and Use Committee (IACUC), approval number U-2022/F/01.

### 2.9. Vaccination Challenge and Sampling

A total of 112 fingerlings were randomly distributed into 7 groups (*n* = 16) according to Table 3. Corresponding to each streptococcal strain there was one vaccinated and one non-vaccinated group. Vaccinated group fingerlings were intraperitoneally (IP) injected with 50 µL of the P3SAB strain vaccine. Two weeks post vaccination, fish were challenged with 100 µL of 1 × 10^5^ CFU/mL of respective *S. iniae* strains, as shown in Table 3.

Fish were monitored continuously post challenge for 14 days for signs of infection, such as unilateral or bilateral exophthalmia, eye opacity, disorientation, loss of equilibrium, hemorrhages at the base of the fins and in internal organs, darkening of the skin, pale livers, and enlarged spleens. Ten days post challenge samples were collected for bacteriology and serum analysis. Relative percent of survival (RPS) was calculated according to the following formula: RPS = 1 − (% mortality in vaccinated/% mortality in control) × 100.

### 2.10. Antibody Response by ELISA

Antibody response post challenge was measured by whole-cell ELISA using sera from experimental fish [27]. Briefly, high-binding 96-well ELISA plates (3590, Costar, Washington, DC, USA) were coated overnight at 4 °C with inactivated P3SAB cells (100 uL/well) resuspended in carbonate-bicarbonate coating buffer (pH 9.6) to an OD600 = 1.0. The next day, plates were washed with PBS containing 0.05% Tween 20 (PBST) before blocking with conjugate buffer (1% Bovine serum albumin in PBST) at 22 °C for 2 h. Diluted sera (1:10) from surviving fish were applied to the wells and incubated at 22 °C for 2 h before washing with PBST. The secondary antibody was an anti-Asian seabass IgM monoclonal antibody (AquaMab F-02, Aquatic Diagnostics, Scotland, UK). Plates were incubated with secondary antibody for 2 h at 22 °C. After washing, Horseradish peroxidase-conjugated tertiary antibody (Goat anti-mouse IgG; A16066, Invitrogen, Waltham, WA, USA) diluted 1:10,000 in conjugate buffer was applied for 1 h. Color was developed for 5 min using 3,3′,5,5′–Tetramethylbenzidine (TMB; T0440, Sigma-Aldrich, St. Louis, MO, USA) and OD was measured at 450 nm with a Hercuvan NS-100 microplate reader. Antisera from 8 fish per vaccine group were analyzed individually and the results were expressed as a mean OD ± standard error.

## 3. Results

### 3.1. Sequence Analysis

The genomic properties of all the strains are consistent with previous assemblies of *S. iniae* strains, including QMA0248 [22], SF1 [29], ISET0901 [30], and ISNO [31]. Detailed properties are shown in Table 2. Circular plots of all genomes are shown in Figure 1a. All strain assemblies contain approximately 2000 coding sequences, 18 rRNAs (6 each of 5S, 16S, and 23S), as well as 68 tRNAs. The number of protein-coding genes is approximately 1500 per strain. PGAP was able to assign a function to most coding regions with only 5–7% of regions annotated as hypothetical proteins, which is consistent and even exceeds in some cases the annotation rates for the high-quality QMA0248 assembly [22]. All assemblies contain the chaperonin-family protein GroEL, which has been shown to be specific to *S. iniae* [32].

To assess the genetic distinctness of our strains we compared the 16S rRNAs with previously reported *S. iniae* strains found in the NCBI nucleotide database. We identified 43 distinct assemblies of *S. iniae* and included the related bacteria *Thermoanaerobacter tengcongensis* (AE008691), *Streptococcus equi* (CP001129), *Streptococcus mutans* (AE014133), and *Lactococcus lactis* (AP018499) in our analysis to gauge the genetic distance within the *S. iniae* species relative to related *Streptococcus* species. All strains of *S. iniae* include multiple copies of 16S rRNA, and this pattern holds true for other *Streptococcus species*. The limited genetic divergence of individual copies compared to the overall divergence within *S. iniae* indicate that these duplication events precede the species formation of *S. iniae*. The resulting phylogeny clearly situates our strains within the *S. iniae* clade, but also shows that the reported strains are distinct from previously sequenced strains of *S. iniae*, as they form a distinct clade within *S. iniae* (Figure 1b).

Orthofinder [25] was then used to categorize protein-coding genes and analyze the homology of protein-coding genes between strains. Of all genes, 99.1% were assigned to orthogroups, and just 20 genes out of a total of 10,370 protein-coding genes across all seven strains could not be associated with orthologous genes in other strains. A total of 1706 orthogroups were identified, with on average about 1250 protein-coding genes shared between each pair of strains. On average, 1240 of these orthologous mappings are one-to-one, indicating on one hand low levels of gene duplication throughout the *S. iniae* genome and at the same time high consistency between strains from geographically different regions, namely Singapore and Australia, as well as across host species. Using the STAG method [27] as implemented in OrthoFinder, we constructed a phylogenetic tree of all seven strains discussed here and nine fully annotated strains available in the NCBI database. We again found that our strains form a distinct subgroup that shares a more distant ancestor with previously described *S. iniae* strains.

The capsular operon plays In important rIlI in immune evasion for *S. iniae*. Hence, we identified the genomic location of the capsular genes and extracted the sequences from our assemblies. Capsular gene locations are indicated by orange blocks in Figure 1a. A detailed view of the capsular operon in the proposed vaccine strain P3SAB is shown in Figure 1d. The capsular genes appear highly conserved throughout all strains, with the sequence identity across the capsular operon exceeding 99.9%, as shown in Figure 1c. In addition, the phylogenetic relations do not coincide with geographic regions or the general phylogenetic patterns observed for the full genome. This appears to indicate a highly stable region that is an effective antigen and hence a good candidate for vaccine-induced immunity.

Manzer et al. [33] identified a number of prominent surface antigens in different species of the *Streptococcus* genus. We extracted the reference sequences for the antigens SpaP (P23504), SspA (Q54185), SspB (Q54186), BspA (Q8E589), AspA (Q48S75), Pas (Q9KW51), SpaA (Q53414), PAaA (Q9LBG3), and PAh (Q59HN9) from the Uniprot database and used these sequences as a blast query against blast databases built on our *S. iniae* strains. A summary of hits for the P3SAB strain is given in Table 4. The results for all strains including the exact E-values are included as a supplementary data file. We were able to identify candidate genes for AspA, BspA, PAh, Pas, SpaA, and SpaP consistently and PAaA in four of the seven genomes. We were unable to identify orthologues to SspA or SspB in any of our strains.

### 3.2. Vaccination and RPS

The fingerlings in both vaccinated and non-vaccinated groups were exposed to respective *S. iniae* strains via IP injections (Table 3), while one group served as negative control and was injected with 0.1 mL PBS. From day 3 onward post challenge, the fish in positive control groups started to show classical *S. iniae* clinical signs with accumulated mortality of 75–100% (P3SAB and Si 6-21 strains showed 0% and CB Si-1 a 25% survival rate). However, all the vaccine groups were healthy with no observed clinical signs, and the survival percentages in these vaccinated fish were above 80% (Figure 2a).

### 3.3. ELISA

In order to determine the cross-protective antibody responses in vaccinated fish, serum was collected and analyzed by whole-cell ELISA. Compared to control group fish, significantly higher serum antibody titers were observed in the vaccinated groups (Figure 2c). Antibody responses were also observed in the non-vaccinated group, which did not provide cross-protection against *S. iniae* infection.

## 4. Discussion

In this study, we characterized seven distinct strains of *S. iniae*, a bacterial pathogen that is highly lethal to Asian seabass. The strains originate from aquaculture operations in Australia and Singapore and thus span a wide geographic area in Southeast Asia and were collected from distinct species. We found that the strains are closely related genetically, with sequence identities around 95%. The capsular genes, which have previously been identified as important surface antigens [18], show minimal variability (Figure 1c). Additionally, we were able to identify several candidate genes previously identified as important antigens in other species of the *Streptococcus* genus [33].

In addition, we show that inoculation with the P3SAB strain was universally protective against various strains of *S. iniae* infection. Survival rates without inoculation were 0% for infection with P3SAB and Si 6-21 and 25% with CB Si-1, consistent with the expected high mortality (Figure 2a). When inoculated with the P3SAB strain vaccine, a significant increase in survival to above 80% against all tested strains was observed. The presence of the serum of antibodies that are reactive with different *S. iniae* strains as shown by ELISA experiments is further evidence of successful vaccination. However, these preliminary data are based on small-scale experimental trials. In the future, additional large-scale trials are required to corroborate the results.

Our work demonstrates that the P3SAB-based inactivated vaccine is clearly effective in protecting populations of *Lates calcarifer* and potentially *E. tetradactylum* from *S. iniae* infections, and thus provides enhanced protection to fish grown in aquaculture. These findings are consistent with earlier findings of Wang et al. [34] demonstrating the effectiveness and cross-protection of tilapia species using formalin-killed-cell-based vaccines based on *S. iniae* and *S. agalacitae*. The vaccine is relatively straightforward and economical to produce and thus holds great promise to improve yields in seabass aquaculture throughout Southeast Asia and Australia. This work also demonstrates a viable strategy for the rapid development of vaccines for new and emerging bacterial pathogens in aquaculture operations. Previous efforts have been made to use protein-based vaccines derived from α-enolase. Liu et al. [35] show that pure recombinant α-enolase shows comparatively low effectiveness against *S. iniae* infection in tilapia, with survival rates of approximately 31%, but demonstrate that these rates can be improved using a carbon-nanotube-based carrier system to about 69%. Wang et al. [36] also demonstrate the effectiveness of α-enolase-based vaccines against *S. iniae* infection in channel catfish, with survival rates of about 45%. These studies provide some indication that effective protein-based vaccines against *S. iniae* are technically viable. Our identification of conserved antigens might enable a future vaccination strategy based on other recombinant proteins or mRNA that might be broadly protective against a variety of related *S. iniae* infections, but it is unclear at this point if such a strategy can be economically employed in aquaculture.

## Figures and Tables

**Figure 1 vaccines-11-01443-f001:**
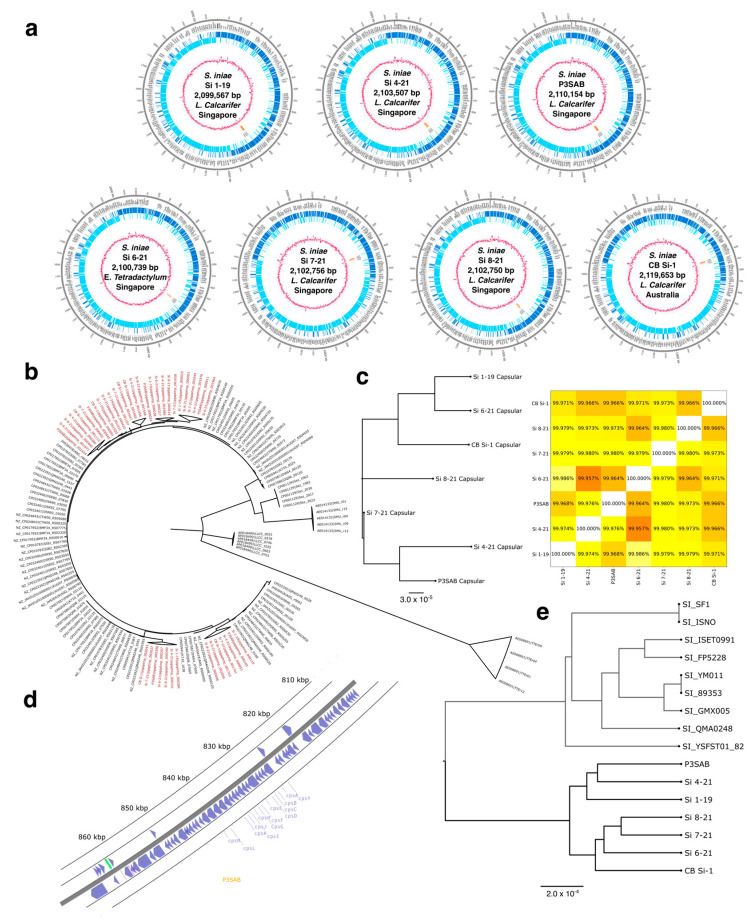
Characterization of 7 *S. iniae* strains. (**a**) Schematic view of *S. iniae* genome assemblies. All samples assembled consistently to circular genomes of 2.1 Mb. Forward-strand genes are indicated in dark blue, reverse-strand located genes are indicated in light blue, and the capsular operon location is indicated in orange. The central magenta line indicates local GC content in a 1 kb window. (**b**) Phylogenetic tree of *S. iniae* and related *Streptococcus* species 16S rRNA. It can be seen that the 7 strains characterized here fit within *S. iniae* but form a distinct clade. (**d**) Detailed view of the capsular operon in the vaccine strain P3SAB. All capsular proteins are present in series. (**e**) Phylogenetic tree derived from protein orthogroups. The geographic association is preserved, showing strains Si 1-19 -P3SAB originating from Singapore as a single clade with low internal genetic diversity, whereas the Australian strains Si 6-21-CB Si-1 show some genetic variation at the protein level. (**c**) Phylogenetic tree and sequence identity matrix of *S. iniae* capsular operon. The sequences of the capsular operon are highly conserved with an average sequence identity of well over 99%. It is evident that the clade segregation is not associated with origin geography, unlike the pattern observed for all proteins. This indicates a high degree of genomic stability in this region and the association of the capsular genes with virality and immunogenicity make these conserved antigenic regions highly attractive for vaccine design.

**Figure 2 vaccines-11-01443-f002:**
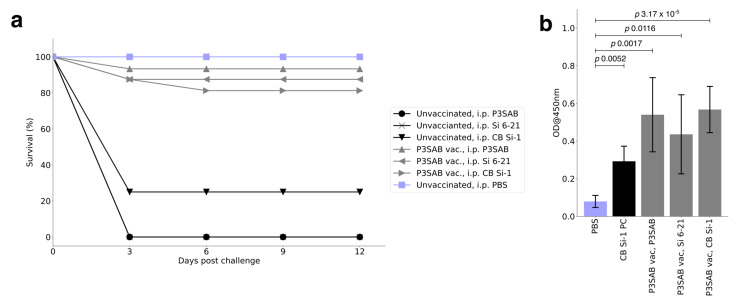
Vaccine efficacy in vivo. (**a**) Percentage survival of control and vaccinated fish during the trial. Overall, unvaccinated groups showed higher mortality percentage (75–100%) compared to vaccinated groups. (**b**) Graph showing serum antibody responses between non-vaccinated and vaccinated fish (mean ± SE). Vaccinated group showed significantly higher antibody titers compared to non-vaccinated groups.

**Table 1 vaccines-11-01443-t001:** Summary of *S. iniae* isolates.

Strains	Origin	Species	Year Isolated
P3SAB	Singapore	Asian seabass	2016
Si 1-19	Singapore	Asian seabass	2019
Si 4-21	Singapore	Asian seabass	2021
Si 7-21	Singapore	Asian seabass	2021
Si 8-21	Singapore	Asian seabass	2021
Si 6-21	Singapore	Threadfin	2021
CB Si-1	Australia	Asian seabass	2018

*S. iniae* strains were isolated on Columbia Agar plates with 5% sheep blood (63784, Bio Rad, Hercules, CA, USA). Aseptically, swab samples were collected from the brain, liver, kidney, and eyeball of the diseased fish showing clinical symptoms. Agar plates were incubated at 30 °C for 48–72 h. Colonies were checked for *Streptococci* morphology (cocci in chains) under microscope.

**Table 2 vaccines-11-01443-t002:** Sequencing, assembly, and annotation data.

Strain ID	Si 1-19	Si 4-21	P3SAB	Si 6-21	Si 7-21	Si 8-21	CB Si-1
Origin	Singapore	Singapore	Singapore	Singapore	Singapore	Singapore	Australia
Accession	CP129326	CP129327	CP129328	CP129329	CP129330	CP129331	CP129332
Host	*Lates calcarifer*	*Lates calcarifer*	*Lates calcarifer*	*E. Tetradactylum*	*Lates calcarifer*	*Lates calcarifer*	*Lates calcarifer*
Sequencing Date	2 September 2021	2 September 2021	2 September 2021	27 December 2021	27 December 2021	27 December 2021	27 December 2021
Total Read length	764 Mb	1063 Mb	808 Mb	3675 Mb	5940 Mb	3754 Mb	2413 Mb
Reads N50	13,145	14,745	15,134	15,820	14,895	15,212	16,608
Reads N90	8188	8575	8368	9480	8832	8755	9937
Assembly Length	2,099,239	2,103,151	2,109,823	2,100,739	2,102,756	2,102,750	2,119,653
Average Coverage	351	492	375	1699	2761	1752	1109
Number of Genes	2013	2021	2033	2074	2086	2088	2092
CDSs	1923	1931	1943	1984	1996	1998	2002
rRNA	18	18	18	18	18	18	18
tRNA	68	68	68	68	68	68	68
Protein-Coding Genes	1524	1546	1531	1445	1448	1471	1476

**Table 3 vaccines-11-01443-t003:** The experimental groups for animal trials.

Groups	Treatment	Vaccine	Challenge Strain	Serum Collected
1	Negative control	PBS	PBS	Yes
2	Group A	*S. iniae* P3SAB	*S. iniae* P3SAB	Yes
3	Group A positive control	PBS	*S. iniae* P3SAB	-
4	Group B	*S. iniae* P3SAB	*S. iniae* Si 6-21	Yes
5	Group B positive control	PBS	*S. iniae* Si 6-21	-
6	Group C	*S. iniae* P3SAB	*S. iniae* CB Si-1	Yes
7	Group C positive control	PBS	*S. iniae* CB Si-1	Yes

Challenge inoculum was prepared by culturing *S. iniae* strains in 50 mL TSB up to an optical density of 0.5 at 600 nm. Cells were washed twice and resuspended in PBS to an OD600 = 1, which corresponds to 1 × 10^8^ CFU/mL.

**Table 4 vaccines-11-01443-t004:** Antigen candidate genes in *S. iniae* strain P3SAB.

Antigen	Gene Locus	Gene Tag	Gene Description
AspA	[1,912,471:1,912,948] (−)	pgaptmp_001861	LPXTG cell wall anchor domain-containing protein
AspA	[1,976,370:1,977,459] (−)	pgaptmp_001909	YSIRK-type signal peptide-containing protein
BspA	[480,510:483,732] (+)	pgaptmp_000491	hypothetical protein
PAh	[155,147:156,713] (+)	pgaptmp_000195	LPXTG cell wall anchor domain-containing protein
PAh	[1,912,471:1,912,948] (−)	pgaptmp_001861	LPXTG cell wall anchor domain-containing protein
Pas	[220,766:221,828] (+)	pgaptmp_000261	HAMP domain-containing histidine kinase
SpaA	[155,147:156,713] (+)	pgaptmp_000195	LPXTG cell wall anchor domain-containing protein
SpaA	[480,510:483,732] (+)	extdb:pgaptmp_000491	hypothetical protein
SpaA	[1,912,471:1,912,948] (−)	extdb:pgaptmp_001861	LPXTG cell wall anchor domain-containing protein
SpaP	[155,147:156,713] (+)	extdb:pgaptmp_000195	LPXTG cell wall anchor domain-containing protein
AspA	[1,912,471:1,912,948] (−)	extdb:pgaptmp_001861	LPXTG cell wall anchor domain-containing protein

## Data Availability

The genome assemblies of all strains are deposited in the NCBI genome database under accession numbers CP129326, CP129327, CP129328, CP129329, CP129330, CP129331, and CP129322.

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
