# Peer review of "Whole Genomic Characterization of Streptococcus iniae Isolates from Barramundi (Lates calcarifer) and Preliminary Evidence of Cross-Protective Immunization"

_vaccines, 2023, doi:10.3390/vaccines11091443_

Round 1
Reviewer 1 Report
This study (manuscript) shows a cross-protective immunization against Streptococcus iniae strains in Lates calcarifer. It is a well-performed study and the results are interesting. Vaccination with the inactivated P3SAB strain of S. iniae showed high cross-protection against both Singaporean and Australian S. iniae strains.
I have just a few minor comments and questions.
Line (L) 49: Clinical signs……includes; change to: Clinical signs…..include (plural)
L 62: A number of proteins were; change to: A number of proteins was (singular)
L 79: Threadfinfish (E. tetradactylum) Abbreviations should be written in full the first time they are mentioned in the text.
L 155: formalin with a final concentration of 0,4% (v/v). Please detail/specify the making of formalin.
L 176+L 181: use superscripts
L 194: BSA, please define
L 198: HRP, please define
Please chech the English grammar mistakes, especially plural/singular cases
Author Response
We thank the reviewer for their favourable comments. We have addressed the issues with the text as below.
Line (L) 49: Clinical signs……includes; change to: Clinical signs…..include (plural)
corrected.
L 62: A number of proteins were; change to: A number of proteins was (singular)
corrected.
L 79: Threadfinfish (E. tetradactylum) Abbreviations should be written in full the first time they are mentioned in the text.
this is first given in full on L66
L 155: formalin with a final concentration of 0,4% (v/v). Please detail/specify the making of formalin.
we added the supplier (252549, Sigma Aldrich)
L 176+L 181: use superscripts
corrected.
L 194: BSA, please define
corrected.
L 198: HRP, please define
corrected.
Reviewer 2 Report
Manuscript vaccines-2542322 describes the passive immunization of barramundi with an adjuvanted Streptococcus iniae vaccine. The subject is extremely interesting and topical, as the authors insert in the paper a series of important ancillary data characterizing the manuscript.
In general, it is a well-written manuscript, easy to understand and which correctly inserts all the useful and corollary notions of the main topic. I have no particular revisions to suggest, as all the chapters are treated comprehensively and correctly.
There are only small corrections throughout the text, such as inserting all scientific names in italics and moving the numbering of bibliographic citations before the period at the end of the sentences.
I remind the authors to correctly insert the relative wording in the Funding paragraph.
The last point to review is in the references where the citations in points 10 and 11 are the same: therefore all the numbering must be reviewed both in the bibliography chapter and in the entire text, paying attention to the correspondence of the new numbering.
For these easily surmountable reasons, in my opinion the work can continue in the publication process after minor revision.
Author Response
We thank the reviewer for his favourable comments. We have corrected the numbering and formatting of the references and printed all scientific names in italics. We have also updated the funding paragraph.